# The Lived Experiences of Female Immigrant Carers in Madrid, Spain: A Phenomenological Study

**DOI:** 10.3390/healthcare11152206

**Published:** 2023-08-04

**Authors:** Montserrat Ruiz López, Noemí Mayoral Gonzalo, Cayetana Ruiz Zaldibar, David Pérez Manchón, Victor Jiménez Díaz-Benito, Juan Pablo Hervás Pérez

**Affiliations:** 1Centro Universitario Saius, Grupo Metrodora Education, Magallanes nº 3, 28004 Madrid, Spain; montserrat.ruiz.lopez@gmail.com; 2Department of Nursing, Faculty of Health, Camilo José Cela University, Urb. Villafranca del Castillo, Calle Castillo de Alarcón, 49, Villanueva de la Cañada, 28692 Madrid, Spain; nmayoral@ucjc.edu (N.M.G.); crzaldibar@ucjc.edu (C.R.Z.); dpmanchon@ucjc.edu (D.P.M.); 3Department of Sport Sciences, Faculty of Physical Activity and Sport Sciences, Universidad Europea de Madrid, 28670 Villaviciosa de Odón, Spain; victorjimenezdb@gmail.com

**Keywords:** ageing, carers, immigration, relations

## Abstract

Background: Immigration and population ageing represent circumstances with important sociocultural and economic repercussions. Methods: A qualitative study using interpretative phenomenological analysis was conducted to understand the daily lives and the work of immigrant women dedicated to caring for older citizens. In-depth interviews and discussion groups were carried out in a group of 40 immigrant carers. Data were analysed via the constant comparative method. Results: Three qualitative themes emerged from the data: ‘difficult lives’, ‘working in the home’, and ‘the vision of the other’. This study highlights the many difficulties encountered by this population. A sense of vulnerability was described while fulfilling their professional duties due to their migrant condition, experiences of gender inequality, and work status. An emotional connection is necessary to provide care, which is impossible in cases of discrimination. Discussion: Administrative regularisation is necessary to improve the quality of immigrant carers’ working conditions.

## 1. Introduction

By the year 2050, the prevalence of people over 65 years of age in the European Union will be 27.5%. Spain, Italy, and Japan will lead this process of ageing, according to the Spanish Social Security Institute for Elderly and Social Services (Instituto de Mayores y Servicios Sociales (IMSERSO)) [1]. Another important aspect announced by the World Health Organization (WHO) is that in 2020, chronic illnesses were responsible for 73% of deaths worldwide [2]. Therefore, although life expectancy is increasing, this is accompanied by processes and side effects that entail a greater need for care.

Traditionally, Spanish families have typically assumed the care of ill or vulnerable family members; however, the informal and/or family-based care system is undergoing a profound change. Grandparents no longer live with their families, as was the case in previous generations. To explain this, Delicado [3] points to the following factors: new family models, lack of availability of carers (mainly caused by the integration of women into the public sphere), as well as limitations related to infrastructure and urbanism. These changing circumstances have resulted in the introduction of the Law of the Promotion of Personal Autonomy and Attention to People in Situations of Dependency in Spain in 2006 [4]. This law recognises the importance of participation and the obligation of the state regarding the provision of care for dependent people as a universal right.

The ageing of the population and the lack of informal carers has contributed to a “pull factor”, especially in the case of women [5,6,7], mostly of Latin American origin [8,9,10,11], given their openness and cultural and language proximity. The care of dependent people (most of whom are older people) by immigrant women majorly contributes to the health status of dependent people [12,13,14] and decreases their discrimination [15,16]. The families and receivers of care prefer them, as they embody a series of qualities linked to the feminine role. In this manner, a transfer of responsibilities has occurred from woman to woman [11,17,18].

Migrant women carry out their work in the privacy of other people’s homes, which can make them vulnerable and essentially invisible to the outside world, as individuals, as workers, and as carers. This results in triple discrimination: the unequal gender relations that exist in our society, the limitation of rights to which migrants are subjected, and, finally, their dedication as carers [7,19,20,21]. In general, migrants are women with an educational level that is frequently higher than that of natives with the same job occupation [22,23,24]. Despite their qualifications and having a regularised legal status, most of them experience difficulties in obtaining recognition of their professional qualifications. This means that finding suitable work at their levels of education is complicated, and, therefore, in most cases, they are relegated to the grey economy, such as domestic carers [15]. In many cases, these people face violations of fundamental labour rights (wages and breaks), underemployment, and a high incidence of contractual informality or seasonality. In these circumstances, they are not protected by accident or sickness insurance, nor can they obtain a financial pension when they reach retirement age [25].

The population profile of immigrant women in Spain includes women aged between 27 and 39 years with children and a medium-to-high level of education. One-third of these women are employed in domestic services, generally as live-in helpers (living and working in the home of the employer). These women frequently consult health professionals due to somatisation because of stress associated with the relocation process, feelings of loneliness, and their difficult social situation [15].

Caring for older people represents a niche in the job market in the community of Madrid due to several factors [12]: an increase in the number of elderly people and people who live on their own (thus requiring external carers), the wishes of older people and their families to remain in their homes, as well as more individual values that lead to demands for attention among the elderly. Immigrant women also have reasons for accessing the care sector [12]: they have a place to stay, meals are secured, and they feel safe in these homes even without the legal paperwork.

Thus, in the case of Spain, two interests have coincided: on the one hand, those of immigrant women who need work, and on the other, those of older people who need care. However, despite this widespread phenomenon, little is known regarding the circumstances of the carers themselves. Therefore, we sought to explore the experiences of immigrant female carers in Spain to learn about their needs and further understand their contributions towards the care of dependent people.

## 2. Materials and Methods

To fulfil the study aim, qualitative methods were chosen. We used the phenomenological interpretative approach proposed by Heidegger to give a voice to immigrant women while considering that their lives are impacted by their histories and cultures [26,27].

### 2.1. Participants

Participants were recruited for this study via convenience sampling. Potential participants (*n* = 40) were identified and contacted at training courses on caring carried out in a non-governmental development organisation (NGO) and at a religious order. The main researcher explained this study’s purpose and gave potential participants a contact sheet to arrange the data collection process.

The only criterion for inclusion was the provision of non-professional care and being migrant women living in the community of Madrid (Spain).

### 2.2. Data Collection

Focus groups and semi-structured interviews were conducted with the study participants. Initially, we performed an exploratory focus group with 6 participants. Subsequently, we performed up to 20 in-depth interviews with immigrant women. The interviews were carried out at an NGO and a religious order in the city of Madrid. For data triangulation, we finally performed 2 further focus groups, with 6 and 8 participants, respectively. The people in the focus groups did not participate in the interviews. At this point, we considered that data saturation was achieved.

### 2.3. Analysis

We used the constant comparison method proposed by Glaser and Strauss for data analysis [28], which consists of generating grounded theory in the data. This method of analysis requires that the data be analysed from the beginning. In this way, we can recognise both what is common and what varies so that questions can be incorporated: what is happening and why is it happening? The method of constant comparison serves as a guide during the analytical process in qualitative studies. It is, therefore, not exclusive to a particular research tradition. It can be used for ethnographic, phenomenological, and grounded theory studies [29]. Codes and fictitious names were given to the informers to anonymise the information. Thereafter, an analysis of the transcriptions of the interviews took place. The Atlas-ti version 6.2. software for qualitative studies was used for the analysis of the data.

### 2.4. Ethics

This study used an uncovered research approach while always respecting the confidentiality of informers. All participants knew that these recorded conversations were part of an uncovered research study.

This research study was reviewed and approved by the Ethical Committee of Camilo José Cela University, Spain.

Each person in this study was anonymised with acronyms and numbers, and only the researchers had access to the recordings and transcripts. Finally, they were presented in the results with pseudonyms to make their voices closer.

## 3. Results

In total, 38 Latin American women and 2 women from Eastern Europe (Spanish speakers) participated in this study. Although the mean age of participants was 41.27 ± 1.23 years, most arrived in Spain when they were young. In general, the participants were educated women, considering that 42.5% had studied at the high school level or higher. Of these women, 16 were married, although 6 of these arrived in Spain as single women. Together with the single women, some of the women were divorced, and thus, 60% of the migrant group under study came to Spain while not depending on male counterparts. Regarding the work performed, most of the women in this study worked as live-in carers (65% of the 40 women in this study).

After the data analysis, the emerging categories and meta-categories were grouped and renamed, obtaining three qualitative themes: difficult lives, working in the home, and the vision of the other (Table 1).

### 3.1. Difficult Lives

This theme encompasses meta-categories that refer to the migratory project, the irregular administrative situation, and the economic precariousness of migrants.

The lives of people who, for whatever reason, have abandoned their countries, their lives, and their families and must begin again are not easy. The “European dream” as described by the carers themselves, is not always fulfilled. They came to Spain to improve their standards of living, only to find that life in the destination country was not what their compatriots had told them, as reflected in the following quote:
“It is not what they had told me, that people here make two or three thousand euros. I swear that in Peru I could work and make eight hundred euros. I wonder how they make the money because then they come to Peru, buy lands, this, that and the other.”(C.17 Monica)

All the study participants described a sense of grief, remembering the families they had left behind. This is especially felt at the end of the day, and at night, when they are alone:
“This is terrible. Only my pillows and I know how many nights I have wept.” (C.8 Julia)

One of the coping strategies used by the carers is the work itself, as it helps them to feel occupied and feel tired and is an outlet to avoid thinking about their families the whole time.

The situation of administrative irregularity of many of these women means that they have no employment rights, which puts them at greater risk of suffering abuse from individuals and experiencing discrimination from society. These women feel the need to legalise their administrative situation, and thus, numerous references to a “work visa” appeared in their discourse.

Among all these difficulties, the prospects of returning to their countries were also dire, due to the uncertainty of the economic future back home. In other cases, their children were studying and had made their new lives in Spain, making it even more difficult to leave. In this sense, the women in this study reported feeling under a lot of pressure, understanding that returning home would be “a failure”:
“And then you come home, and you feel terrible because of all the work your parents have gone to. Going back home is like being a failure and then all is lost.” (C.14 Nuria)

Another participant said the following:
“All the people at the beginning are a little difficult, they are nervous because they don’t know us. The employee is not only an employee, but she is also a friend. The employee becomes a confidant, she is a friend, she is the person with whom you spend a lot of time, and of course, since you arrive new and another girl has been there before, the person is nervous. The first three months are very hard. You must start treating them, and it is very important to give them affection. You find her nervous, screaming, and irritated. Then you talk to her with a lot of respect and affection. Then everything goes well. You must try to talk to her and be quiet when she wants to be quiet. You must share things with her. For example, she likes Cine de Barrio, so I watch it with her as if I liked it, and since she can’t hear or see well, I tell her what’s going on. There is a rejection until you put them in your pocket. As the colleague says, we manage.” (GDC.2 Margarita)

### 3.2. Working in the Home

This theme describes the perceived characteristics for which these women are chosen to work as carers in people’s homes. It also describes how the long days go by within the employers’ homes.

Latin American women are preferred by Spanish families and are more frequently hired than women or men of other nationalities. According to the women interviewed, a decisive factor to explain this is that they charge less than native women:
“Because it is cheaper for them [laughs] and because, in our situation, we must do the work for less. Do you really think that for what I am earning a Spanish woman would work as a live-in carer? No way!” (C.12 Sonia)

Most women were live-in carers, which was described as a job that comes with advantages and inconveniences. Working and living in someone’s home is important, especially for women in irregular administrative situations. It provides them with a sense of security as they need not worry about arrangements for settling into the new country, nor do they have to worry about the dangers of walking around a city that they do not know. However, despite the sense of security and savings that come with this job, some of the women acknowledged feeling like they were being “locked in”:
“I never go out. I am always looking out for the family, if I want to go out to do some shopping, I must call the niece. Being locked in and not being able to go out for a walk kills me.” (C.17 Mónica)

The barrier between the personal and work environment becomes blurred in the case of live-in carers. The participants described feeling under the control of their employers regarding their space and food:
“[…], when you are an external carer, you know that you are going to go out. For example, I am going to go to my room, sit down, eat calmly, drink my Coca-Cola relaxed. However, there, you can’t do that. Sometimes they are overlooking to see whether you drank the milk or whether you left the TV on, or they even enter your room.” (C.7 Ana)

The lack of work regularisation is reflected in this study, reflected above all in the constant complaint of a lack of rest time:
“I cried, and cried, I mean, Saturday is like a day off, but if someone or another comes, you don’t have a timetable.” (C.19 Lucia)

### 3.3. The Vision of the Other

This theme refers both to the relation with the persons under their care, as well as their contexts. The relationships they develop are marked both by the work they perform and their condition as migrants. Taking care of someone means face-to-face contact. Over time, relationship bonds are strengthened between the older people and their carers: “togetherness brings affection”. According to the carers, some of the people cared for perceived the great value of the care they received and demonstrated this by showing gratitude, protection, and support in specific situations. However, carers also spoke of the inability to develop good relationships when they are made to feel discriminated against for their origins. Sometimes, they remembered circumstances where their employers addressed them in a way that showed clear signs of racism:
“She said that us, Peruvians, are useless and that we are unable to do greater things, which is why we have the work we have. She always marginalised me, ‘you have grown-up next to animals who are unable to do more, that’s why you come here, there are lots of your compatriots doing the work you do’. When I told her that I was studying geriatrics, she said, that’s good, that’s a work for immigrants, you can study to take care of children or cook.” (C.5 Miriam)

## 4. Discussion

### 4.1. Difficult Lives

The situation of long-term care in Spain, specifically in the 2006 Law for the Promotion of Personal Autonomy and Care for Dependent People in Spain [4], generates challenges that create inequality in access to services. In addition, the economic crisis affecting Spain in recent years has strongly aggravated the situation for our informers, which adds to the harshness of the migratory process itself [30,31,32]. Faced with this situation, domestic care work is seen as an opportunity in difficult times.

This study highlighted that the situation of administrative irregularity in which many of these women find themselves constitutes an important source of stress. Trying to obtain a “work visa” is a hard task involving considerable effort. This administrative situation also turns them into vulnerable subjects, prone to suffering abuses by native residents, their own compatriots, and mafia networks. These findings do not correlate with those of previous studies, which mainly emphasise the benefit of migratory networks [17,33].

Thus, this study found classic problems relating to the entire migratory process. Indeed, we found references to the seven immigration losses defined by the psychiatrist Achotegui [34], although the most frequent complaints concerned family loss, followed by loss of status. According to how the migratory experience develops, the effects of the separation from the country of origin (family, culture, etc.) end up being endurable. When things go well, the pain and sadness, or even depression, subside; on the other hand, when things worsen, the depression/stress duality is, as affirmed by Achotegui [34], a balance between what is gained and what is lost. These women can also feel themselves in a state of loss due to the different cultures. They miss the food (tastes and preferences), the climate, the light, the colours, and the warm temperatures typical of their home countries. Previous studies regarding Latin American immigrant workers describe cases of women suffering from physical disorders, such as back pain, digestion problems, and headaches [35]. Other studies describe these women as being very depressed and having problems sleeping, which is in line with our findings [36]. Indeed, sleeping disorders are a common finding in the case of Latin American carers, who are reported to suffer from insomnia because of being away from their families [37]. The role of carers could be enhanced if working conditions were to be improved with the aim to increase the desire of individuals and families to give more importance to long-term care at home versus care provided in institutions [38].

### 4.2. Working in the Home

Although, from the onset, being a live-in carer is a guarantee of a place to live and food, plus providing a possibility of becoming legal, in general, the aim of the migrant women who participated in this study was to find jobs to enable them to live separately. The women in this study working as live-in carers faced long working days as they were required to be on call twenty-four hours a day. Furthermore, they lacked their own space, and the employers frequently invaded their space, as well as marking the territory that they considered appropriate for a domestic worker. These caretakers, like family carers, need physical and psychological rest but find this extremely difficult to fulfil. The special characteristics of their work mean that they do not have respite care (families usually hire only one carer). This situation represents a great burden, especially when the older person suffers from acute pathologies and when a sensory processing disorder exists. These findings are supported by a study by Gallart [32], showing that immigrant carers suffer a burden like that of family carers, as well as being at a greater risk of suffering from fatigue, stress, and depression because of their job.

Several factors appear to favour the hiring of Latin American women for domestic care jobs and, more concretely, for the provision of care for dependent people. These are mainly three factors: women in this group are considered experts in caring as they come from traditional societies (where care for the elderly and/or sick people is provided by the families themselves), they possess the advantage of having mastery of the language, and their fees are lower than those of their native counterparts.

Carers also carry out household maintenance, as well as supervising all the daily living activities. To improve their conditions, the Spanish government approved the Royal Decree-Law 16/2022 for the improvement of Social Security working conditions for workers in household services that guarantees their skills and recognises occupational diseases and bonuses [39]. However, they are the link between dependent people and the formal healthcare system, administering the treatments prescribed by doctors and accompanying them to the hospital when needed. But, above all, they provide supervision and companionship, keeping them active and preventing their dependency from worsening. Most carers come to this job without experience, learning via trial and error. They consider that training is important because it would help them to increase the quality of the work they perform. They must also familiarise themselves with the character of the elderly, who due to functional limitations, pain, and loneliness, take out their frustrations on their carers, so they must be very patient to get along with them. Various analyses of long-term care policies in Spain have been carried out, including of the figure of social workers, but currently, there are important deficiencies in social services, and interventions are insufficient or poorly directed [40]. In this context, it is essential that the public administration reinforce support and consolidation with women’s associations, since their level of proximity is effective.

The great responsibility and being on call all day, and sometimes at night, means that care is a job that requires presence and dedication. Carers feel tired because of the long days without rest, but what is really exhausting for them is when they cannot sleep because the elderly person’s condition worsens.

### 4.3. The Vision of the Other

The relationships that are established are marked by triple discrimination: as a woman, an immigrant, and a domestic worker [7]. Female carers felt that their employers were taking advantage of the legal, economic, and cultural constraints associated with their situation. To overcome this common scenario, considerable effort is involved, as the carer must demonstrate her competence to gain a degree of complicity and appreciation from the person under their care and the family. This depends mainly on whom the employer is and the characteristics of the dependent person, their perception regarding immigrants, and the possible prejudices and stereotypes. The carer may or may not overcome the initial distrust that prevails to eventually succeed in “winning over” the affection of the family, as well as achieving desirable work recognition.

The native population are perceived as having a closed character with a notable emotional detachment, and which clashes with the carers’ open and chatty natures. Indeed, adapting to Spain and to their employers can lead to feelings of strangeness, inferiority, ignorance, and fear [37]. Several women found that they were treated with contempt, in general, by the Spanish and their employers. These findings add to previous research describing food discrimination (immigrant women are often given smaller portions, forced to eat in the kitchen, and given lower-quality food) [41,42]. Among the testimonies gathered in this study, there were only a few instances of rejection and discrimination that were openly and directly racist. More often, the informers spoke of events or certain ways the employers talked to them that represented covert racism. This reaffirms the idea that traditional racist discourse based on biology has been replaced with a more subtle and elaborate discourse, which some authors define as a new method of racism, based not so much on skin colour but rather on culture: this is a type of “everyday racism” [43]. Despite this, and given that daily care generates closeness and cohabitation, the relational value of care means that on many occasions, prejudices are overcome, and relationships are established that are based on mutual respect [11,37,44,45]. Addressing these challenges will likely require comprehensive reforms to improve the efficiency and effectiveness of the long-term care system in Spain. In the immediate future, the government of Spain will develop training and accreditation policies for domestic and care workers to analyse the conditions of internal workers [46].

## 5. Conclusions

The migratory project is full of commitment, hope, and the promise of money. Thus, immigration represents a complex social phenomenon, which includes political, economic, social, and political factors that condition the entire migratory project. It is necessary to recognise the valuable contribution that immigrant carers make to Spanish society, as they are covering a demand that neither the formal system of care nor families are covering.

Holistic caregiving implies that, in addition to manual skills, certain qualities such as observation, gentleness, and empathy are needed. Many of these qualities were present in the women in this study, but they needed to learn how to care for the dependent persons. This is why the women carers in our study also demand specialised training.

To improve the provision of care performed by these women, changes of a political and social nature are required to help regulate their work situation. Only by acquiring citizenship rights will they be able to take their work out of the invisibility in which they stand. Furthermore, this may lead to an improvement in the quality of care by increasing potential training possibilities for this population. Ultimately, both public administrations and society at large must know immigrant carers are an important aid in cases of dependency.

The relational and emotional components of care, as in the nursing profession, are difficult to objectify. Nurses demand recognition of the invisible part of care. In the case of migrant carers, we could speak of “double invisibility” as they are not formal carers and are not family members, so recognition is limited to the private sphere and/or to the goodwill of employers.

### New Contribution to the Literature

This study identified the difficulties experienced by a group of immigrant carers in Spain. In particular, the case of female live-in carers represents a delicate situation as these women suffer a lack of intimacy and appropriate rest and are therefore at risk of depression and stress, both of which may affect their health and the quality of care provided.

## Figures and Tables

**Table 1 healthcare-11-02206-t001:** Meta-categories and categories associated with the vector.

Meta-Category	Category
Difficult lives	The illusion of the migration project
	Irregular administrative status
	Migratory mourning
The home as a place of work	Positive discrimination: gender, ethnicity, and price
	Spaces are not one’s own
	Carer’s overload
	The emotional component of care: responsibility and patience
The view of the other	Triple discrimination
	Prejudices and stereotypes
	Relationships of mutual respect

## Data Availability

The data presented in this study are available upon request from the corresponding author. The data is not publicly available because the transcribed interviews are protected by Organic Law 3/2018, of 5 December, on the Protection of Personal Data and guarantee of digital rights.

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
