# Peer review of "The Lived Experiences of Female Immigrant Carers in Madrid, Spain: A Phenomenological Study"

_healthcare, 2023, doi:10.3390/healthcare11152206_

Round 1
Reviewer 1 Report
Dear authors, thanks for giving me the oppportunity to review this paper. It covers a very important topic and fills a gap in terms of research of live-in carers in Spain.
The paper aims to describe the lived-in experience of female immigrant live-in carers in Madrid.
The paper would benefit from a few more linkages with the literature on live-in carers (e.g. Brigitte Aulenbacher, Michael Leiblfinger, Almut Bachinger - does not necessarily have to be these authors, but there is literature out there on the experiences of live-in care).
further comments:
Abstract: interviews and discussion groups were 'carried out' rather than 'performed on'
Introduction:
the 'open nature' - do you mean perceived open nature?
'Many' families and receivers of care prefer them - not sure if all
they embody a series of qualities - what are the qualities?
relegated to 'private jobs' - the grey economy?
in Spain is a women - woman
2. Method
carried out vs performed, see above
uncovered research approach - please explain
3. Results
please locate the interviewees geographically - were they all working in Madrid?
'not depending on male counterparts' or 'not living in couple relationships?'
most ....worked as live-in carers - what were the others working as?
After the data analysis - please add coding process in more detail
3.1. which makes them at greater risk - which puts them at greater risk
suffering abuse from the society - suffering abuse from individuals, experiencing discriminiation from society
uncertainty the economic future - uncertainty of the economic future
3.2. the characteristics for which these women - the perceived characteristics for which these women
Itr provides them....- ? no settling in and walking around a city just because they are live-in carers?
3.3. 'togetherness brings affection' - source =?
employees addressed - employers addressed
4.1.
abuses by the autochthonous - what about those who are neither autochthonous nor compatriots? are they all in the mafia?
4.2.
they come from traditional societies - explaion why this is a factor, what is associated with a traditional society (e.g. gender roles)
4.3.
they were treated negatively - what is negative treatment?
5. Conclusions
regularisation as leading to better care vs. declared aim to change jobs mentioned in an earlier section - please link this 8only some?)
English is quite good, only small corrections, see above among general comments
Author Response
All the changes we have made, as well as all the new text and corrections to the text, are in the template in red.
Changes realized:
- The abstract was revised and corrected
- Introduction were made as the reviewer suggested
- Methods were made as the reviewer suggested
- Results were made as the reviewer suggested (also in 3.1, 3.2 and 3.3)
- Discussion were made as the reviewer suggested (also in 4.1 (change autochtonous by native, 4.2 and 4.3)
- The conclussions were update

Reviewer 2 Report
First of all, I apologize for the delay in my feedbacK regarding this article.
The article meets the necessary requirements for publication, the introduction and contextualization is adequate, the selected methodology is perfectly adapted to the objectives set out in the study. The presentation and explanation of the categories found is interesting. Therefore, from my point of view, it meets the minimum requirements to become a publication.
congratulations.
Author Response
All the changes we have made, as well as all the new text and corrections to the text, are in the template in red.
The article was revised by us, adding more information and clarifying the errors that were detected during the review.

Reviewer 3 Report
Thank you for submitting your manuscript on the lived experience of female immigrant carers in Madrid Spain. This is a fascinating topic and a much needed area of research. You identify some very important themes that represent the challenges faced by these immigrant workers. I was curious as to how you came to choose the constant-comparative method for data analysis as this is frameowork generally more compatible with a grounded theory approach than a Heideggarian phenomenological approach to inquiry. Can you provide examples of how data was coded and units of data compared to one another? Also how were auditability, credibility, and fittingness addressed (criteria for addressing the validity of qualitative data). Were participants allowed to review your thematic analysis and descriptions and confirm whether these accurately captured their lived experiences? You mention that 65% of your sample were live-in carers? What was the nature of work of the other 35% of participants? Were there differences in themes based on whether participants were live-in workers? Also, I noticed that the reference list is quite dated with most references over 10 years old. Can you update your references to within five years? Thank you again for your submission.
This manuscript was very difficult to read due to the quality of the English language. There were also words I was not familiar with or words and phrases that were unclear. I would suggest the authors clarify the meanings of a term that is not likely to be understood widely by the reading audience ("autochthonous").
Author Response
All the changes we have made, as well as all the new text and corrections to the text, are in the template in red.
The article was revised in English to ensure that it was written correctly.

Reviewer 4 Report
In the Introduction, describe the work situation for immigrant women who are carers. Are there any protections? Why is it considered precarious work in the Spanish context? More background on carer work (e.g., pay, benefits, legal rights, etc) will assist with understanding the Results and quotes under your themes.
Some Introduction sections that were particularly confusing follow:
Page 2: The ageing of the population and the lack of informal carers have contributed to-wards a “call effect”, especially in the case of women [5-7], mostly of Latin American origin [8-11] due to their open nature and command of Spanish. The care of dependent people (most of whom are older people) by immigrant women represents a positive de-velopment in cases of dependency [12-14] and serves to decrease discrimination towards the same [15,16]. The families and receivers of care prefer them, as they embody a series of qualities linked to the feminine role. In this manner, a transfer of responsibilities has occurred from woman to woman [11,17,18].
Feedback: The section above is not clear to me. What is the “call effect?” I need more clarification of how this concept is related to dependency, discrimination and family preferences. Although there are references associated with these concepts, more description is needed, especially a definition of call effect and its evidence-based links to the other concepts mentioned in this paragraph.
Page 2: Migrant women perform their work in the intimacy of other people’s homes, which can make them vulnerable and essentially invisible to the outside world, as individuals, as workers and as carers. This results in a three-fold discrimination: against women, im-migrants, and domestic workers7. In general, migrants are women with an educational level that is frequently higher than that of autochthonous women who work the same jobs [19-21]. However, in most cases they do not enter the market according to their qualifica-tions as, although they may be in a regularized legal situation, they have difficulties hav-ing their qualifications officially recognized. This means that finding suitable work at their level of education is complicated and, therefore, in most cases they are relegated to private jobs, such as domestic care [15].
Feedback: In the paragraph above, as written, it’s not clear how there is three-fold discrimination arising from women working in carer roles. I recommend defining each type of discrimination and providing an example and reference of each type. Or, you can save this for your Discussion where you discuss triple discrimination.
Under Methods, you need to explain your purposeful selection of participants. What were your inclusion/exclusion criteria? Why did you intentionally select 2 women from Eastern Europe?
Under Results, some tables would be helpful, such as a demographics table for participants and a table of your themes/categories and sub-themes/categories to provide an overview of your key thematic findings. I’m not sure what you mean by meta-category. A table with levels of your themes or categories, therefore, would provide a visual representation of your results.
For each category, theme, you need to define themes and sub-themes. For example, what was your conceptual definition for “difficult lives” and the three themes associated with it? There should be quotations associated with the sub-themes under “difficult lives.”
I recommend looking at another qualitative paper published in the journal to get an idea of organizational formatting and how to highlight your key findings.
I liked the Discussion. There were nice links between the themes and the existing literature. See my comments above about triple discrimination. Discussion is where you can interpret your findings and link to other literature. Overall, I think you did this in your Discussion.
Overall, the Introduction needs to provide more background on the carer role and why you consider it precarious work in the Spanish context. The methodology is briefly explained and there are no clear inclusion/exclusion criteria.
More details and definitions are needs of themes and sub-themes in the Results. Tables might be used for participant demographics and the themes or categories you found.
I liked your Discussion. What would add to your paper is political insights of what needs to happen to bolster the role of carers at different society levels (government, communities, in homes, for example). As written, the paper is very short and needs more context in each section of the paper.
There were some paragraphs that were hard to follow. The links between concepts were not clear. There were incomplete sentences and some tense issues.
Author Response
All the changes we have made, as well as all the new text and corrections to the text, are in the template in red.
Changes realized:
- The abstract was revised and corrected
- Introduction were revised and changed as the reviewer suggested
- Methods were revised and changed as the reviewer suggested
- Results were revised and changed as the reviewer suggested
- Discussion were revised and changed as the reviewer suggested
- The conclussions were update
The article was revised in English to ensure that it was written correctly.

Round 2
Reviewer 3 Report
Thank you for submitting your revisions to the manuscript. I do believe that it is a stronger manuscript as a result of your attention to reviewer comments.
This new version is a bit easier to follow and the quality of English does seem to have improved.
Author Response
The authors really appreciate this comment. Thanks for this comment. You are totally right. Following your comment, we have carried out a thorough revision of the English language to bring our article to the optimum quality required by the reviewer.
We have carried out a modification of the different parts of the article (you can find them in red for the modifications of round 1, and in blue for the modifications of round 2). These modifications made to our article, we believe, make the article even clearer and make it easier for the reader to read.

Reviewer 4 Report
This is an important topic, and I appreciate the changes made. The readability is much improved with useful details added. I have one request:
The Discussion usually does not include additional data from the Findings. There is a quote included in the Discussion that fits better in your Findings section.
I found a few awkward terms, such as "uncovered research..." Please read through one more time to ensure all terms are relatable to readers.
Author Response

(The authors gave the same response as above.)
